# The Role of Virtual Reality as a Psychological Intervention for Mental Health Disturbances during the COVID-19 Pandemic: A Narrative Review

**DOI:** 10.3390/ijerph19042390

**Published:** 2022-02-18

**Authors:** Muhammad Hizri Hatta, Hatta Sidi, Shalisah Sharip, Srijit Das, Suriati Mohamed Saini

**Affiliations:** 1Department of Psychiatry, Universiti Kebangsaan Malaysia Medical Centre (UKMMC), Cheras, Kuala Lumpur 56000, Malaysia; hizhatt@gmail.com (M.H.H.); hattasidi@hotmail.com (H.S.); shalisah@ppukm.ukm.edu.my (S.S.); 2Department of Human & Clinical Anatomy, College of Medicine & Health Sciences, Sultan Qaboos University, Al Khoud, Muscat 123, Oman; s.das@squ.edu.om

**Keywords:** COVID-19, mental health problems, digital devices, virtual reality applications

## Abstract

The COVID-19 pandemic spread throughout the world and created many problems. The COVID-19 pandemic caused an increase in mortality and morbidity, including mental health problems. Around the world, the movement control order (MCO) was strictly enforced, but the spread of the infection epidemic was still rampant. The magnitude of the increase in mental health illnesses has caused many individuals to suffer. Given that face-to-face interventions are challenging to carry out during an outbreak, we need to address this critical problem through an online approach, such as virtual reality (VR). This approach is vital to helping patients deal with their existing problems in more pragmatic, practical, and customer-friendly ways. Thus, in the present review, we proposed the development of a virtual digital device for this noble purpose. Various challenges, improvements, and expectations for VR applications were outlined and discussed in this narrative review.

## 1. Introduction

Several individuals with an unexplained cause of pneumonia were discovered in Wuhan, Hubei Province, China, in December 2019 [1]. The World Health Organization (WHO) termed this viral infection spreading in lung pneumonia as COVID-19. The pandemic has spread quickly and rapidly to more than 200 countries, with more than 71 million verified cases. More than 1.6 million deaths were reported on December 16, 2020 [2]. The COVID-19 pandemic has caused not only mortality from the infection but also horrendous and severe adverse psychological sequelae for people all over the world [3]. This high-intensity stress response is thought to be a root cause of a variety of poor physical and psychological health effects [4]. 

These adverse psychological effects encompass a wide range of cognitive-emotional and bodily reactions. Such reactions include anxiety, depression, a decrease in the function of the immune system, and insomnia (initial, middle, and terminal), which have surfaced since the pandemic [5,6]. In a systematic review published in 2020, a range of factors was associated with a higher risk of psychiatric sequelae and/or low mental well-being, including female gender, deteriorating self-related physical health, and families with COVID-19 [7]. 

Discovering the methods or strategies of directing the public to effectively and appropriately control their emotions amid public health catastrophes has thus become an urgent concern for those seeking to prevent and avoid losses caused by this massive crisis. [8].

In Malaysia, the restriction or movement control order (MCO) was imposed with a call to “stay home, stay safe” by the government from March 2020 to August 2021. The MCO banned residents from close contact, i.e., socializing and organizing events, meetings, and gatherings. As a result of the continuous restriction order, psychological and emotional suffering such as stress (or distress), anxiety and depressive symptoms, anger, petulance, and emotional burnout has been reported throughout the COVID-19 pandemic [9]. Several psychological consequences have been recorded in the general Malaysian community as a result of the MCO, including an increase in the prevalence of anxiety, depressive, and stress symptoms [10]. Higher levels of negative emotions were also reported among the young (less than 35 years old) individuals, who faced the pandemic, and these included stress, fear, boredom, insomnia, and irritability [10].

To relieve psychological tension, a variety of stress management and relaxation strategies is utilized, and online psychological intervention may be one of them. From this perspective, a simple intervention, such as the attention restoration theory, asserts that people can focus better after paying more attention to nature or glancing at nature photographs. It states that one may reduce stress by exploring nature [11]. As a result, the virtual reality (VR) experience in this kind of setting may be a stress reliever [12]. A pleasing natural scenery and a peaceful ecological setting, for example, can significantly relieve and repair human functions [13]. 

The instruction to ‘remain home, keep safe’ restricts and limits an individual’s outdoor movements because of the ongoing spread of COVID-19. People could not access natural beauty and attractions comfortably. Researchers observed that VR could reduce stress and improve a person’s mood if presented with a scenery of isolated and confined environments [14]. Indeed, as early as 2001, many medical and health researchers and physicians used VR technology to improve the health of the individual [15]. VR is used as a diversion technique to help patients cope with their stress. [16,17].

## 2. COVID-19 Pandemic, Stress, and Mental Health Disturbances

Stress refers to positive and negative cognitive, emotional, behavioral, and biological responses to a perceived threat [18,19,20]. Stress is also a common life sphere for the indicators for the measurement of quality of life (QoL) [21]. It gauges well-being [22]. A study suggested that one way of managing stress is by taking time out to do activities in which one has already attained well-being [23]. 

During the movement lockdown imposed by the government, psychological and emotional sequelae, such as stress, depression, and anxiety, rose among people because of the intrusion into day-to-day activities. The predisposing determinants for poor mental health are poor financial support and decreased social interactions, among others. The presence of insomnia, inadequate physical and recreational activities, and having a relative or contact with COVID-19 are correlated with a mental health issue. Moreover, the fear of being infected and receiving negative information regarding the spread of the pandemic were also added as risk factors for psychological problems. Other psychological factors, e.g., past childhood misfortune and stressful life experiences and poor coping styles, were also important. The protective factors against adverse mental health include residing with a family, having great social support from peers, and assertive family functioning [8,24,25,26,27,28,29,30]. 

A systematic review and meta-analysis reported that maintaining a physical distance of 1 m or more would provide the quantitative standards for COVID-19 spread prevention [31]. These findings emphasized the optimum face mask usage and protection of the eye in public and healthcare contexts. A sound randomized experiment is needed in the future to consolidate evidence for these interventions better. This methodical evaluation of the best available data may support the importance of social distance.

The purpose of this article was to conduct a narrative review on the role VR as a psychological intervention for mental health disturbances during the COVID-19 pandemic. Our research questions are whether VR is beneficial for the psychological intervention approach in patients with mental health problems. 

In the first part of our review, we articulate the effect of the pandemic on an individual, which causes a significant effect of psychological distress. Due to the magnitude and prolonged period of the pandemic, the delivery of the mental health system needs to be pragmatic and adapt to a non-face-to-face setting, with digital health technologies playing a role in mitigating the sufferings of people with mental health problems. We discussed the importance of VR as a strategy to help people in distress during this pandemic. This includes the benefits of the immersion, interactive, and affordance of VR, and the direction of the discussion also focuses on the challenges encountered with VR technology, such as developing a specific setting for therapy, affordability, and accessibility. Finally, we would conclude that our review was based on the studies that we narrated.

## 3. The Use of Computer-Based Technology as a Psychological Intervention: The Role of VR

Since there is an importance for social distancing, the best intervention should be a non-face-to-face setting such as an online psychological intervention embedded in VR. VR refers to: “a computer-generated environment with scenes and objects that appear to be real, making the user feel they are immersed in their surroundings” [32]. A person perceives this environment through a medium or tool, which is known as a VR headset, gadget, or helmet. We can immerse ourselves in an interactive environment as if we were learning how to intervene in a specific situation via VR. Although it appears to be very futuristic, its inventions are not as cutting-edge as we might think. The Sensorama, a machine with a built-in seat that controlled a 3D visual scenario, gave off scents, and produced vibrations to make the experience as realistic as possible, is widely regarded as one of the first VR experiences. The discovery dates back to the mid-1950s. Over the years, the subsequent technological and software advancements have gradually evolved in devices and interface form and design.

An earlier study planned and built VR as a two-dimensional experience (self-location and possible action) [33]. This device serves as a platform to promote positive states of emotion, such as enjoyment. Enjoyment is a mental, emotional, and cognitive state that leads to engaging in a formative activity [34]. Involvement is the concept of a psychological state stemming from the focus and attention given to an activity [35]. Enjoyment and involvement are positive cognitive-affective-motivational states [36] that can be augmented by VR, enhancing these positive states. A recent study reasoned that a heightened sense of presence eases engagement with awareness—the visualized sensations [37].

## 4. Who Is Using VR? An Online Survey on the Characteristics of the Respondents

Based on an earlier study, there was a considerable link between VR access and gender, with more males having access to the VR equipment [38]. Despite this, no significant relationship was found between VR access and age, console access and age, or gender. Among respondents with VR access, the majority felt their usage had increased throughout MCO. This situation helped to keep them busy and occupied. This situation was true for all genders and age categories.

Researchers offered a fresh perspective on how recreational VR use can successfully reduce the negative effects of lockdown times on the emotional and physical well-being of the community [38]. Under the constraints imposed by the lockdown, they reported that VR activities assist users to stay occupied and physically active. Due to their lower cost, technological availability, and accessibility, virtual reality headsets and gear have become standard entertainment and communication devices in many families [39]. As a result, academics, policymakers, and healthcare personnel should examine virtual reality (VR) as a possible public health benefit in intervention techniques to alleviate the detrimental effects of prolonged lockdown periods. Providing the community with opportunities to participate in virtual reality projects in order to keep them engaged and physically healthy could be a potential strategy for reducing the mental and physical health declines that have been noted in many people since the start of COVID-19 [39]. 

This sort of self-administered intervention—VR—could conceivably ease the current burden on healthcare workers (HCW) and social support staff [40]. Nevertheless, a cost/impact examination and analysis would be warranted to evaluate the usefulness, practicability, and effectiveness of replacing or supplementing the current "in-person support strategies" with ad hoc VR-based applications. Researchers provide novel and substantial evidence establishing the usefulness of VR activities to improve the population’s psychological and physical health [38]. However, the survey’s self-reported nature is known to carry inherent risks of response bias [41]. Additional investigations would be advisable to monitor critical physiological and psychological physical and mental well-being indicators under controlled stipulations.

## 5. A Research Model for the Usefulness of VR as an Intervention: The Role of Sense of Presence Guided Imagery, Enjoyment, and Involvement

The conceptual growth of the research model alluded to in prior works was demonstrated [42]. The research model is mostly built on the basis of a few models [33,43,44] to link the concept of presence with an affective-motivational state, a VR experience, and psychological well-being. Figure 1 shows the four-layer theoretical framework for its potential usefulness and benefits in mental health. It found the relevant conceptual basis for the positive effect, such as the affective-motivational status of a VR intervention. Researchers reported that a higher sense of presence in VR journalism leads to greater enjoyment and involvement [45]. Researchers found that a person exhibits involvement and entertainment when using a VR game to learn, specifically an objective, i.e., a game destination in digital interaction [46]. Hence, even if it is not in a real-world, but in the real digital virtual world, a person may experience being present. This experience is associated with the sense of enjoyment and involvement of a virtual environment when a person is watching VR (Figure 2). Yang et al., 2021 investigated the research framework, which was based on a model with the following hypotheses [42]: 

## 6. VR vs. 2-Dimension Intervention for Psychological Stress during the COVID-19 Pandemic

In the midst of the current worldwide pandemic, the need for remote delivery of online mental health interventions, including meditation assistance, has become critical. When interventions are remitted remotely, however, the aid that one might normally feel in the presence of an instructor may be reduced. This scenario could have an effect on one’s meditative experiences. Using head-mounted displays (HMD) to display video-recorded guidance may enhance one’s sense of psychological presence with the expert compared to the presentation via a regular flat screen, such as on a computer laptop monitor. Researchers evaluated a didactic, trauma-informed care approach to instruction in mindfulness meditation [47]. This action is carried out by comparing meditative reactions to an instructor-guided meditation when addressed face-to-face vs. prerecorded 360° movies viewed on a standard flat screen monitor (i.e., 2D format) or via an HMD (the VR headset; 3D format).

In a prior study, young adults (*n* = 82) from a university preparatory course participated in a preparatory course and encountered a 360° video-guided meditation via HMD (VR condition, 3D format) [47]. The respondents were randomly instructed to practice similar meditation techniques via scripted face-to-face guidance (in vivo [IV] format) or observe them on a standard laptop display, i.e., non-VR condition, in 2D form. The respondents reported positive and negative experiences with affective and meditative approaches using a self-reported rating. The meditation breath attention scores (MBAS) were recorded by the researchers during each session. In this study, meditating in VR (3D format) was related to an increased experience of minor embarrassment than in face-to-face instruction. Respondents found VR meditation to be less enjoyable and more tiring. When compared to 2D meditations, VR meditations were associated with more amazing relaxation experiences, less distractibility from the breathing process, and less weariness. The study found no differences between VR and non-VR meditation in focus/concentration (MBAS). Of those who reported choosing one design or configuration, nearly half favored the VR format, and almost half preferred the IV format. Recorded 360° video education in meditation viewed through an HMD (VR/3D format) appears to provide a considerable experiential benefit over teachings presented in 2D design and may provide a safe—and for some, even preferred—alternative to teaching meditation face-to-face.

## 7. VR Development for Psychological Intervention: Technological Innovation, Assessment, and Troubleshooting

Currently, conventional VR systems generate and create realistic images, characters, voices, and other sensations that replicate and simulate a user’s bodily presence in a virtual environment using either VR headsets or multi-projected settings. An individual using VR equipment can navigate the simulated or artificial world, i.e., move in three dimensions, such as walking, moving, or running around in the artificial environment. This action involves interaction with virtual features or objects. The VR headsets created this outcome, which consists of a head-mounted display with a small tube screen in front of the eyes. Still, they can also be developed throughout specifically designed openings with multiple large screens. VR typically consolidates auditory sensation and video feedback but may also enable other sensory modalities and force feedback through haptic technological innovations. 

The main problem with current computer-based psychological intervention is the transformation of rigid-based one-way into an interactive session. These strategies involved technological innovation, assessment, and software troubleshooting and re-assessment, as shown in Figure 3. 

Regarding technological innovation in VR development, issues related to three-dimensional (3D) visualization, dynamic exploration, the trip and flow, immersive experiences, and visual interaction need to be considered by researchers, therapists, and content developers to suit the role of VR for psychological intervention. VR offers an immersive sensory encounter that digitally simulates a virtual atmosphere. Dynamic exploration refers to an experience such as “being able to walk around objects freely, and being able to move around and see them from wide angles”. 3D visualization refers to the three-dimensional experiences offered by the app with interactive visualization, something that is lacking in the 2D textbook guide for mantra psychological treatment. Flow refers to an autotelic experience where “action and awareness merge”. There is great focus and attention on the task, and we pay little attention to “time or self”. Inducing the flow state is a significant affordance of VR. Affordance refers to the ability or role of an object to perform a certain action. It conceptualizes visual elements (or features) of the environment as information [48]. Affordance in VR is a pragmatic concept that allows the participants and the environment to interact to induce the flow state (i.e., subjectivity, perception, roles, context, and plays) in people. A passive user does not feel stimulated by the information but instead allows the person to take an active role in noticing and utilizing the information. Affordances are thus elements of the environment that enable action on the part of those who understand and recognize them. 

VR is a technology that, at least anecdotally, transfers participants into immersive environments, encouraging flow. However, virtual reality raises usability issues that may stifle flow. The novelty of the technological innovation lies in the unique and helpful way that VR can adapt the psychological intervention as the patients are in the actual therapy situation. 

Software difficulties and practical issues such as user-friendly gadget handling, navigation flexibility, and clarity of interactive sessions are pivotal in software troubleshooting. In this troubleshooting, we need to address the technical problems such as improving the process and dealing with crashes during assessment and intervention. 

Technological innovation and software troubleshooting assessment (and reassessment after troubleshooting) are vital to the ideal VR software gadget used for psychological intervention. We need to look at the usefulness and practicality of VR. The effectiveness, which allows a variety of scenario-based therapies, would be gratifying for both the therapist and the patient. A sense of enjoyment and an enriching experience would foster more compliance with this mode of intervention.

## 8. VR Design-Based Research (DBR) for Psychological Intervention for Mental Health: The Evolution of Software Development

Researchers also revealed the findings of a virtual 3D anatomy assessment and viewpoints for educational purposes, which is a notion that is quite similar to VR software development for mental health intervention [49]. We are considering making VR software using the design-based research (DBR) technique, which includes steps including the design, implementation, and evaluation of a virtual 3D assessment scenario. (Figure 4). Consequently, a researcher would redesign, reimplement, and re-evaluate, again and again, to achieve the best-fitted interactive program in the future for psychological intervention.

DBR intends to improve psychological intervention techniques through iterative analysis during the product’s conception, development, and implementation [50,51]. In other fields, such as higher education and medical intervention, the DBR methodology has been widely adopted [52,53,54,55]. Researchers designed and developed mixed reality simulations for skills development in the health and paramedical fields [56]. This process and the evaluation of the actual practice in the psychological intervention are complex. It has been suggested that a thematic analysis was used to investigate the patient responses involved during coding and theme development. This process development is deductive using the existing conceptual framework, including the total experience (practicality, effectiveness, and user-friendliness) and sense of enjoyment. Technical innovation (3D image representation and dynamic exploration) and software troubleshooting (usefulness and technical issues) should be investigated: design and redesign, evaluate and re-evaluate, implement and reimplement, until the finest of software development is improved, testable, and marketable. 

## 9. How to Create a Scenario? The Interventional Settings

The scenarios that are suggested to be incorporated into the VR software development would be of numerous options. We can have a temporary situation wherein the respondent views a chaotic pandemic situation. The respondents wearing the VR helmet and gadget would gauge their biological responses (blood pressure, pulse, and respiration). An anxious or distressed person typically experiences a higher level of pulse rate, respiration, and blood pressure. The blood pressure, pulse, and respiration would be monitored after each succession of psychological interventions. For example, in each session, a therapist asks the respondents to perform a simple bio-feedback response. In this exercise, a directive in the form of a specific mantra, “please breath slowly and deeply until you feel a sense of calmness”, is followed by the respondents immediately rechecking their pulse after the relaxation technique in the immersive and interactive digital platform. Suppose the respondent has pulses at the same or higher rate than their previous rate. They need to repeat the exercise until the mantra of breathing deeply and slowly works by reducing their pulse from the baseline of higher levels of anxiety (or distressed) states to a lower rate. Besides the pulse rate, the respondents may also monitor their blood pressure and respiration rate on the headset screen connected with the auxiliary gadget such as an oximeter and blood pressure monitor. 

Based on a protocol developed by Riva et al., 2020 [57], instituting a natural setting environment such as a secret garden would be another emotionally inspiring strategy to help psychologically distressed people. The Secret Garden is a 360-degree VR scenario where participants are engaged in a naturalistic setting and are in a “safe haven” on a digital platform. They felt calm and relaxed, which is far from the stressful circumstances experienced in routine everyday contexts. Gradually, they may learn how to control their distressing feelings and reflect on their experiences in a guided protocol. In another study by Alyan et al., 2021, a participant who was exposed to stimuli derived from greenery scenarios in virtual forest environments showed beneficial stress-relieving effects [58]. This is based on the outcomes of two physiological techniques that determine their heart rate and skin conductance levels. The beneficial effects would be profound if realistic graphics were to be used. This exposure contributes to the concept of forest therapy and provides new directions for future forest therapy research. For the role of immersive VR therapy in relieving stress symptoms among people with a physical disorder such as chronic obstructive airway disease, Rutkowski et al., 2021 reported benefits in terms of the mood improvement role among patients with lung disease [59]. 

We can also use psychometric tools to measure the outcome, pre- and post-intervention. In this set, only the respondent is present in the setting. Another option is an avatar (representing self), where it interacts with other avatars (representing other distressed people or with the therapist). However, this approach is expensive, due to the involvement of a third party, despite its benefits in virtual group therapy. With third-party participation, i.e., with multiple avatars, we have to wait for other avatars to enter the session and engage within the online platform we are interacting with. Where time is a barrier to interacting with multiple avatars during immersion, this situation is regarded as expensive to run. Another example would be the involvement of numerous therapists (psychologists, psychiatrists, counselors, or trainers in psychological intervention) who need special training to monitor or conduct the session, which ultimately adds to the operational cost.

There are various approaches to measuring the intervention benefits scientifically. One of the simple ways is the visual analog (VAS), which is better than other measurements [60,61]. Another version of the psychometric tool uses a validated psychometric tool, such as the Depression, Anxiety, and Stress Scale-21 items (DASS-21) [62,63]. The DASS-21 is a set of three self-reported scales that are used to objectively assess depression, anxiety, and stress. Dysphoria, hopelessness, devaluation of life, self-deprecation, lack of interest/involvement, and anxiety are among the items on the three-domain DASS-21 scales. The DASS-21 is based on a multidimensional rather than a binary understanding of psychological pathology.

VR in the medical and healthcare setting has grown in recent years. VR serves as an intervention for a calming effect and as a distraction measure for distress and pain. Mindfulness, relaxation techniques, and psychotherapy are all viable additional delivery methods for evidence-based therapies. The introduction of this innovation into a hybrid medicine-rehabilitation inpatient unit may potentially lead to a highly positive reaction in a near-future study. This arrangement also applies to psychiatry and the mental health field. The complexity of the COVID-19 illness and the need for self-isolation during this pandemic pose many barriers. Isolation, lack of privacy, restrictions on physical movement, and a lack of variety in people’s environments and lifestyles were among the pandemic’s major psychosocial challenges.

For this reason, VR served as an escape, while many others believed that it was a coping apparatus or gadget. VR serves as a supplementary modality for the delivery of experiences that would be otherwise difficult to obtain. As an example of an intervention that was employed during COVID-19 (Figure 5), Figure 5 illustrates a VR intervention for distressed staff and patients in a COVID-19 recovery unit (CRU) hospital in New York, NY, USA.

## 10. Implications for Expanding the Use of VR in Communities Experiencing COVID-19 Mental Health Disparities

VR has advanced significantly in the healthcare field in recent years, functioning as an instructional tool, a pain distraction measure, and an extra platform for delivering evidence-based interventions such as mindfulness in neurorehabilitation and psychotherapy. The current study in the literature found that implementing this innovation in a hybrid medicine–rehabilitation inpatient unit resulted in a very positive reaction [42,64]. 

The digital health intervention could provide positive and recovery-focused content that is used with other support options, i.e., psycho-education and face-to-face psychotherapy or psychiatric consultation (with a strict standard operating procedure). In a study by Berry et al., 2019 [65], the participants found digital health interventions acceptable due to the empowering nature of the program, which is tailored for a self-managing capability to take ownership of their own psychological needs. However, there was a concern regarding privacy and confidentiality. There are fears regarding the possibility of health digital/gadgets replacing other mental health service delivery methods.

In other studies by Li et al., 2014 [66] and Fleming et al., 2017 [67], some participants suggested gamification strategies, which have shown promise in improving engagement with a digital platform interaction between users and providers. These strategies are associated with improvements in engagement and motivation to use digital health interventions. This is helpful for severe mental health problems. These questions warrant continued investigation.

Table 1 illustrates the characteristics of the studies with a list of information on the names of studies, year of studies, country of origin, inclusive and exclusive criteria, outcome measures, interventions, and the benefits/disadvantages of VR interventions [47,57,58,59,64,65]. 

The complexity of the COVID-19 sickness, as well as the isolation that comes with it during a pandemic, poses significant obstacles. Many patients on this ward stayed in the hospital for more than three months, and visitors were almost entirely barred from the facility between late March and late June 2020, in accordance with the New York State Department of Health requirements. Isolation, confinement, and a lack of variety in the surroundings and lifestyles of patients were also key psychosocial problems throughout the pandemic. VR acted as an escape for some participants in the study, but it also worked as a coping skill for others. VR can be used to give experiences that would otherwise be difficult to access.

During the pandemic, the capacity to quickly execute the VR program on an acute COVID-19 recovery facility or on patients who are upset at home and require psychological intervention is crucial. For this noble purpose, it is essential to work closely with the relevant stakeholders, such as psychiatrists, psychologists, information technology experts, and hospital staff.

VR intervention is well matched with the COVID-19 preventive plan, despite heightened procedures to avoid infection and maintain patient and provider safety. Response bias or observer–anticipation effects are downsides of the evaluation approach, which are mostly attributable to staffing shortages during the pandemic.

As patients’ treatment strategies and plans are formulated according to this approach, we should know the multiple providers. This concern for prejudice and preference is far from the only treatment option available in the healthcare paradigm nor is it the only treatment option that plays an exclusive role in patient care. The following steps in gathering future feedback on this approach could be through an inspirational therapist, trained research assistants, or dedicated Ph.D. and master students. With advancements in technology, it is crucial to help and treat patients with mental health problems during this pandemic (Figure 6).

## 11. Limitations

Admittedly, this review has a few limitations. A narrative approach often does not meet the essential criteria to mitigate bias. Not uncommonly, they lack unambiguous criteria for article selection. A narrative review is the lowest form of review possible and is subject to extensive bias in the selection of articles. In this narrative review, we included qualitative studies, cross-sectional studies, and randomized clinical trials and summarized the findings in Table 1. A systematic review would fill the gap in the specialty of VR for psychological intervention among individuals with mental health problems, as the narrative review has potential bias and the search for the studies is not systematic and methodologically reproducible. As another point, there is always a limit to the barriers to VR technology. Although it is accessible, that does not mean that it is affordable and reachable for all users. In other words, VR is a luxury that is not available to everyone.

## 12. Conclusions

In summary, ideally, VR should consist of a fully immersive, 3D ecosystem that transports people to participate in interactive environments, thereby promoting positive psychological well-being. VR technology can assist in the training, evaluation, delivery, and supervision of psychotherapy skills. Patients reported positive psychological effects, particularly in a naturalistic setting and when they were in control of their bodily symptoms. This strategy makes VR a valuable tool for mental health intervention and treatment. On the other hand, VR exposure therapy (VRET) enables individualized, steady, controlled, immersive exposure that is simple for therapists to employ for treatment. The VR approach is often more suitable for patients than in vivo or imaginal experiences. Subsequently, the continued collaboration and partnership between developers, researchers, psychiatrists, psychologists, and patients are critical to advancing the processes of VR software development for psychological interventions. 

## Figures and Tables

**Figure 1 ijerph-19-02390-f001:**
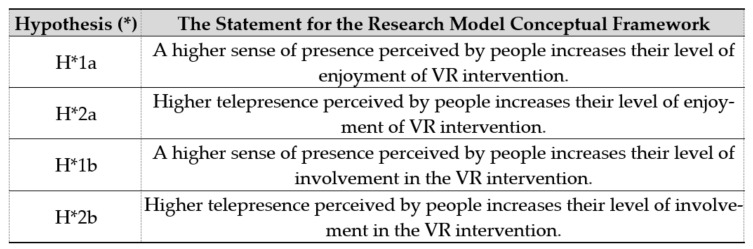
Figure shows the four-layer theoretical framework of a potential Virtual Reality (VR) intervention for mental health. This concept evolves from a study by Yang et al., (2021) [42]. Based on Yang et al.’s perspective [42], we speculate on the interventional study of VR. The first is the concept of presence (or telepresence). The second is the mediation by VR. The third is the affective-cognitive-motivational state of the participants. Lastly, it reduces stress, anxiety, depression, and other mental-health-related problems resulting from the intervention.

**Figure 2 ijerph-19-02390-f002:**
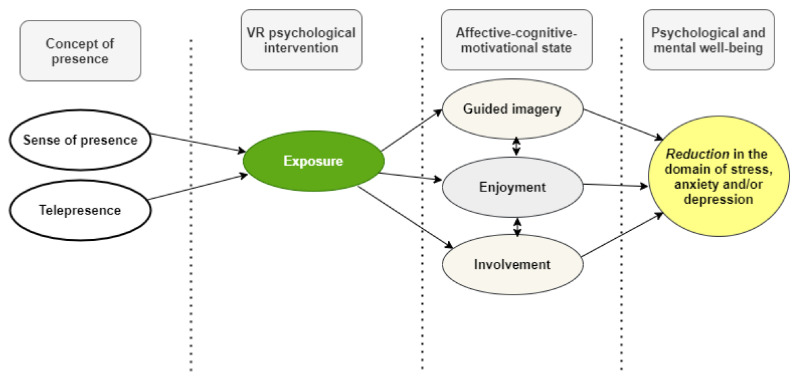
The Virtual Reality (VR) software development. The themes and subthemes in the schematic map for VR development and assessment.

**Figure 3 ijerph-19-02390-f003:**
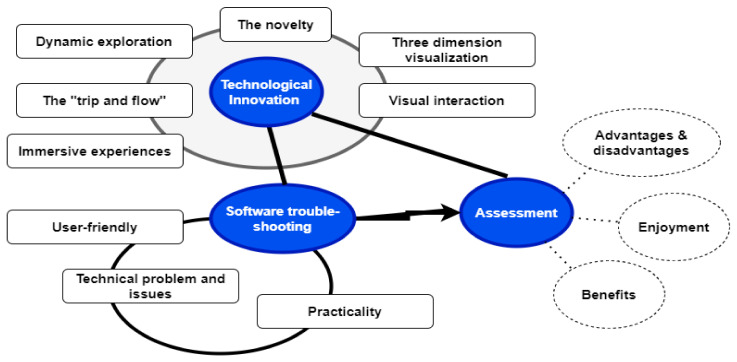
Schematic diagram of themes and subthemes illustrating the map of potential VR software development as part of VR psychological intervention for mental health problems. This software development involves numerous stakeholders, namely: patients, therapists (psychiatrists and psychologists), and software developers.

**Figure 4 ijerph-19-02390-f004:**
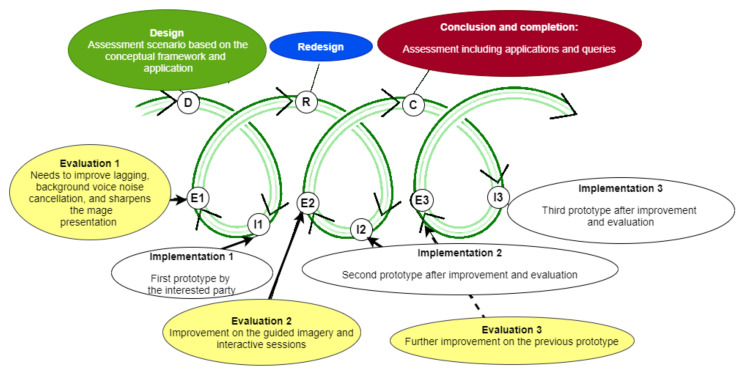
An example of design-based research (DBR) for a VR psychological intervention for patients with mental health problems. This model is a trial-and-error endeavor to refine and use the best software program to intervene in psychological distress and possible psychiatric disorder. It consisted of a design, first evaluation, implementation, and subsequent evaluation. Subsequently, another process occurs until we achieve the best VR software with user-friendly, cost-effective, and therapeutic benefits.

**Figure 5 ijerph-19-02390-f005:**
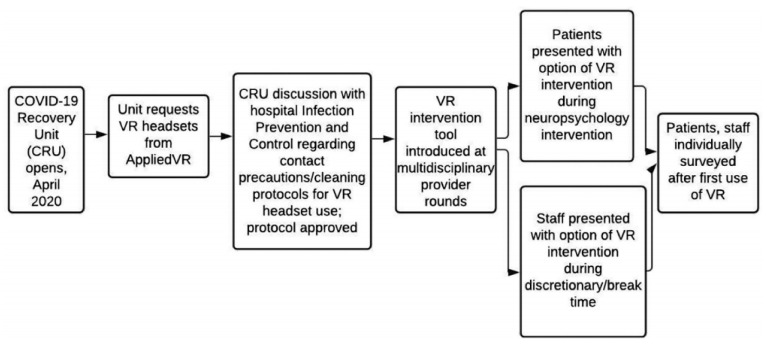
A flowchart describing the implementation of the VR tool within COVID-19 in a COVID-19 recovery unit (CRU). Reprinted from Kolbe et al., (2021) [64], Copyright (2022) with permission of Elsevier.

**Figure 6 ijerph-19-02390-f006:**
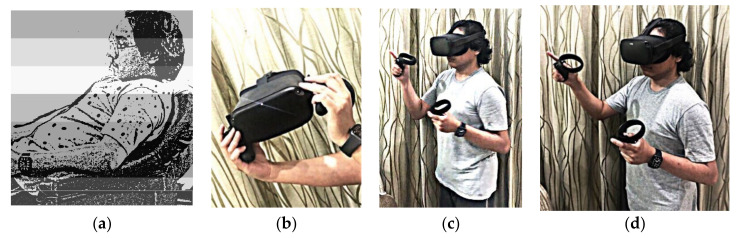
(**a**) (extreme left) shows a man with a mental health problem. (**b**) (second from left) shows a gadget of a VR Oculus for psychological intervention. (**c**) (second from right) and (**d**) (extreme right) display how a VR Oculus is used during the intervention.

**Table 1 ijerph-19-02390-t001:** Characteristics of the studies.

Study	Study Design	Country	Setting	Inclusion Criteria	Exclusion Criteria	Intervention	Exposure Measurement Scale	Outcome Measurement	Comparator/Control	Statistics (E.G., OR/RR, *p*-Value, 95% CI)	VR-Based Intervention (Outcome)
Waller 2021 [47]	RCT	Canada	Not defined	Not defined	Not controlled for, but participants evaluated for life events, childhood events, traumatic events, PTSD, and life experiences before commencing study	Non-VR group (1) traditional face-to-face (in vivo [IV] method), (2) pre-recorded 360° video viewed by standard laptop computer monitor (2D format), and (3) prerecorded 360° video viewed through an HMD (VR condition; 3D format)	A modified emotional questionnaire, Buddhist Affective States, Meditation Breath Attention Scores Meditative Experience Questionnaire	Experiences of relaxation, less distractibility from the process of breathing, and less fatigue	3D (VR) vs. 2D format	Qualitative thematic analysis	When compared to the 2D format, VR meditations were associated with more significant
Riva 2020 [57]	Multicentric, pragmatic pilot randomized controlled trial	Italy	Online	Adult patients (≥18 years); Of the mother tongue of the country where they will be enrolled; Have experienced at least two months of quarantine or isolation related to the coronavirus pandemic; Give full, written, informed consent; Have the availability of a smartphone and a Cardboard VR headset; Availability and agreement of a partner for conducting the self-help component of the treatment.	Visual or ear impairments that can limit the participation in the protocol. Participants reporting vestibular and/or balance disorders.	The 10-min “Secret Garden” 360° VR experience	Perceived Stress Scale (PSS) Depression Anxiety Stress Scale (DASS-21) Beck Hopelessness Scale (BHS) Social Connectedness Scale (SCS)	A reduction in anxiety, depression, perceived stress, and hopelessness, as measured by DASS-21, PSS, and BHS. A reduction in state anxiety and subjective distress, as measured by SUDS. An increase in relaxation, as measured by SRSI3.	Control (waiting list) vs. Two-Group Random Assignment Pretest–Post-test Design	N/A	No intervention (at a protocol level)
Alyan 2021 [58]	Cross-sectional	Malaysia	Online	Healthy university students	Eye impairment	VR intervention with a forest environment, i.e., one realistic experience (RE) and the other dream-like state experience (DE)	Physiological Index, i.e., the heart rate (HR) and skin conductance level (SCL) Physiological measure, i.e., Profile of Mood States (POMS) questionnaire	Relaxation in the domain of the psychological index and low HR and better SCL	Healthy control in RE and DE	Two analytical methods were used: (1) for the HR and SCL data before and after the LDT, related paired *t*-tests were carried out to verify whether the LDT played a role in increasing stress levels, and (2) for the same indicators (HR and SCL), the differences b	The use of VR led to significant decreases in participants’ psychological and physiological stress
Rutkowski 2021 [59]	RCT	Poland	Pulmonary rehabilitation conducted in a ward setting	Patients with chronic obstructive pulmonary disease (COPD), age 45–85 years; anxiety or depressive symptom score of >8 on the Hospital Anxiety and Depression Scale (HADS)	Cognitive impairment; inability to self-complete the research questionnaires, presence of disturbances of consciousness, psychotic symptoms or other serious psychiatric disorders at the time of examination or in the medical data; initiation of psychiatric treatment during the research project; contraindications for VR therapy (epilepsy, vertigo, eyesight impairment)	A VR TierOne device (Stolgraf^®^, Stanowice, Poland) as the VR source. A head-mounted display with total immersion created an intensely visual, auditory, and kinesthetic stimulation	Perception of Stress Questionnaire (PSQ), Depression and Anxiety Depression Scale (DASS), Evaluation of Functional Capacity (EFC)	The changes in stress levels and depressive and anxiety symptoms was the primary outcome. As a secondary outcome, we evaluated functional capacity.	Immersive VR therapy and the control group performed 10 sessions of Schultz autogenic training	Effect size between control and experimental group using Shapiro–Wilk test, the Mann–Whitney U test and repeated-measures analysis of variance (ANOVA) [68]	↑
Berry 2019 [65]	Qualitative study (interviews)	United Kingdom	People with severe mental health problems focus on two domains: (1) views about Digital Health Interventions (DHIs) for severe mental health problems, and (2) ideas for future DHI content and design features.	(1) Diagnosis of a schizophrenia-spectrum disorder or bipolar disorder; (2) 18–65 years of age; (3) Capacity to provide informed consent; (4) Sufficient English language skills; (5) Internet and mobile phone access.	Recruitment stopped when data sufficiency was reached; that is, based on analysis of transcripts and discussion amongst the research team, it was agreed that no additional themes were generated from the data	Digital health interventions (DHIs)	Thematic analysisbased on the role of VR intervention	Self-empowerment of VR use Considerations must be made about who has access to DHI data and how it is used	Nil	Data were analyzed thematically	↑
Kolbe 2021 [64]	Cross-sectional	USA	CRU (COCID-19 Rehabilitation Unit)	(1) Inpatients with positive COVID-19 PCR test during hospitalization (2) Medical team deems patient medically stable and has ongoing medical and rehabilitative needs (3) Able to tolerate >30+ min PT/OT each daily (4) PT or OT recommendation for Acute or Subacute rehab at the time of discharge (5) Anticipation of remaining in hospital/rehab for ≥ 1 week (6) No active SI, Severe dementia and active delirium, (7) Must have noninvasive O2 needs of 6 L or fewer, or in case of tracheostomypatients have achieved “trach collaring” with anticipated ability to downsize/decannulate.	N/A	VR with maximum use time by AppliedVR is 30 min	Simple 1–10 yes/no rating scale (10 indicates the highest satisfaction and highest recommendation)	Satisfaction (“Feeling of enjoyment and get connected with what they see”) Perceived enhancement (“excellent escape and immersive experience”)	Patients and staff	13/13 patients answered “yes” to recommending the therapy to others, and 12/13 answered “yes” to the perceived enhancement of their treatment. 11/11 staff answered “yes” to recommend the therapy to others, and 11/11 answered “yes” to perceived enhancement of their well-being	↑

↑ = Increased benefit and advantages; VR = virtual reality; OR = odds ratio; RR = relative risk; 95% CI = 95% confidence interval; β = beta statistics; PLS-SEM = partial least squares structural equation modeling; LDT = letter-detection test; PT = physical therapy; OT = occupational therapy; and SI = sexual intercourse.

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
