# Peer review of "The Role of Virtual Reality as a Psychological Intervention for Mental Health Disturbances during the COVID-19 Pandemic: A Narrative Review"

_ijerph, 2022, doi:10.3390/ijerph19042390_

Round 1

Reviewer 1 Report

Thank you for the opportunity to review this narrative on the role of virtual reality and software development as a psychological intervention for mental health disturbances during the COVID-19 pandemic.

The title is slightly confusing as virtual reality is dependent on software development, so how is software development a psychological intervention. The paper describes how any virtual reality, in general, is beneficial to mental health. It may be that the authors are suggesting that specific mental health software is required. If so, the title should reflect this, as in “The role of virtual reality and mental health software development …”

A problem with narrative reviews is that the authors are not required to report their search strategy, inclusion and exclusion criteria, or the databases searched. This makes the process unreproducible and introduces the risk of author bias.

It is not clear if the authors have developed software, line 251, or if they have developed and used software 298-304 as there is no reference cited for the “current study”.

The first part of the paper reports on the mental health benefits of virtual reality, in general, with the reduction of stress, anxiety and depression. This is supported by papers, over and above those cited, reporting the use of VR during the pandemic. While the authors appear to want to develop software, hence the section on software development, they do not make a strong case for the need to develop mental health-specific virtual reality interventions.

Section 3, lines 93-97: avoiding face-to-face intervention may be best during the Covid-19 pandemic, but the implication is that virtual reality is the best intervention. No data are presented to suggest that it is better than a videoconference intervention with a psychiatrist or psychologist. The data presented in Section 6 suggest that there is no specific preference between in vivo and virtual reality.

In section 4: does the information presented refer to mental health-specific virtual reality activities or interventions, or for example, virtual reality games? For example, do lines 134-137 refer to mental health-specific interventions or any use of virtual reality?

Figure 3: while it is stated later that there should be stakeholder engagement in software design and development, this is not brought out in figure 3.

Section 7, line 232: the concept of affordance should be explained.

Section 9: how will the respondents gauge their biological responses?

Section 9, lines 282-285: why is it expensive? The involvement of the third party and the associated cost needs to be explained.

Figure 5: what was the nature of the implanted virtual reality tool? Was it mental health-specific?

Section 10: there are no references to support any of the statements made.

Section 10: what is meant by “virtual reality arbitration”. No reference is provided.

Author Response

Reviewer.1

Thank you for the opportunity to review this narrative on the role of virtual reality and software development as a psychological intervention for mental health disturbances during the COVID-19 pandemic.

The title is slightly confusing as virtual reality is dependent on software development, so how is software development a psychological intervention. The paper describes how any virtual reality, in general, is beneficial to mental health. It may be that the authors are suggesting that specific mental health software is required. If so, the title should reflect this, as in "The role of virtual reality and mental health software development …"

We note your concern regarding the confusing title, and we have revised the title to make it unambiguous. The new title is:

The Role of Virtual Reality as a Psychological Intervention for Mental Health Disturbances during COVID-19 Pandemic: A Narrative Review

A problem with narrative reviews is that the authors are not required to report their search strategy, inclusion and exclusion criteria, or the databases searched. This makes the process unreproducible and introduces the risk of author bias.

We thank the reviewer for the comment. We are aware that narrative reviews are unreproducible and risk author bias, but we have included our citations. We also addressed the limitation in a new subheading just prior to the conclusion. We wrote:

"11. Limitations

This review has a few limitations. A narrative approach often does not meet the essential criteria to mitigate bias. Not uncommonly, they lack unambiguous criteria for article selection. However, in our review, we included studies from a peer-reviewed journal, and thus, the credibility and validity of the studies were fair to good. We admitted a systematic review would fill the gap in the specialty of VR for psychological intervention among individuals with mental health problems.

On another point, there will always be a limit to the barriers to VR technology. Although it is accessible, that does not mean that it is affordable and reachable for all users. In other words, VR is a luxury that is not available to all people."

It is not clear if the authors have developed software, line 251, or if they have developed and used software 298-304 as there is no reference cited for the "current study".

We refer to a "current study" as "a near-future study." We intend to develop the software as part of the team's Ph.D. program. In the next two years, we want to build a software program for psychological intervention among patients with mental health problems, and the first author (M.H.H.) is currently doing his Master's in Medical Science, which later will be converted to a Ph.D.

We re-structure the sentences in the same paragraph as: "The introduction of this innovation into a hybrid medicine-rehabilitation inpatient unit resulted in a highly positive reaction in a near-future study."

The first part of the paper reports on the mental health benefits of virtual reality, in general, with the reduction of stress, anxiety and depression. This is supported by papers, over and above those cited, reporting the use of VR during the pandemic. While the authors appear to want to develop software, hence the section on software development, they do not make a strong case for the need to develop mental health-specific virtual reality interventions.

We have included three relevant references to support developing mental health-specific virtual reality interventions.

We wrote: "The digital health intervention could provide positive and recovery-focused content that is used with other support options, i.e.,psycho-education and face-to-face psychotherapy/psychiatric consultation (with a strict standard operating procedure). In a study by Berry et al. (2019), the participants found digital health interventions acceptable due to the empowering nature of the program, which is tailored for a self-managing capability to take ownership of their own psychological needs. However, there was a concern regarding privacy and confidentiality. There are fears regarding the possibility of health digital/gadgets replacing other mental health service delivery methods.

In other studies by Li  J et al. (2014) and Fleming TM et al. (2017), some participants suggested gamification strategies, which have shown promise in improving engagement with a digital platform interaction between users and providers. These strategies are associated with improvements in engagement and motivation to use digital health interventions. This is how it is helpful in severe mental health problems. These questions warrant continued investigation."

Berry, N., Lobban, F. & Bucci, S. A qualitative exploration of service user views about using digital health interventions for self-management in severe mental health problems. BMC Psychiatry 19, 35 (2019). https://doi.org/10.1186/s12888-018-1979-1

Li J, Theng Y-L, Foo S. Game-Based Digital Interventions for Depression Therapy: A Systematic Review and Meta-Analysis. Cyberpsychology Behav Soc Netw. 2014;17(8):519-27.

Fleming TM, Bavin L, Stasiak K, Hermansson-Webb E, Merry SN, Cheek C, et al. Serious Games and Gamification for Mental Health: Current Status and Promising Directions. Front Psychiatry. 2017;7.

Section 3, lines 93-97: avoiding face-to-face intervention may be best during the Covid-19 pandemic, but the implication is that virtual reality is the best intervention. No data are presented to suggest that it is better than a videoconference intervention with a psychiatrist or psychologist.

Thank you for bringing up this issue. As stated in reference no. 46, and in the text (second last paragraph in section 6), we wrote: " Recorded 360° video education in meditation viewed through an HMD (VR/3D format) appears to provide a considerable experiential benefit over teachings presented in 2D design and may provide a safe—and for some, even preferred—alternative to teaching meditation face-to-face." This study by Waller et al. (2021) clearly states that the 3-D format communication, which is the fundamental concept of VR (i.e., being immersive and interactive), is more favourable in meditation teaching than the 2-D format. Videoconference intervention with the psychiatrist or psychologist lacks immersion and self-empowerment to conduct the desirable therapeutical sessions at their convenient time. Also, in a videoconference, the interaction time is minimal due to constraints like limited usability (as a digital platform like Zoom or GoogleMeet has its own terms and conditions).

The data presented in Section 6 suggest that there is no specific preference between in vivo and virtual reality.

Yes, at the current juncture, we are not sure what kind of specific preference would suit the best for therapeutic interventions (in vivo vs. virtual reality). Due to the pandemic and fear of spreading the COVID-19 virus, to our knowledge, the best option setting for both patients and therapists would be a virtual reality intervention

In section 4: does the information presented refer to mental health-specific virtual reality activities or interventions, or for example, virtual reality games? For example, do lines 134-137 refer to mental health-specific interventions or any use of virtual reality?

We elaborated virtual realities activities such as specific interaction for relaxation techniques in your next question on how the users gauge their biological response. Under section 9, we wrote:

“An anxious or distressed person typically experiences a higher level of pulse rate, respiration, and blood pressure. The blood pressure, pulse, and respiration will be monitored after each succession of psychological interventions. For example, in each session, a therapist will ask the respondents to perform a simple bio-feedback response. In this exercise, a directive in the form of a specific mantra, "please breath slowly and deeply until you feel a sense of calmness," was immediately followed by the respondents’ rechecking of their pulse after the relaxation technique in the immersive and interactive digital platform. Suppose the respondent has the pulses at the same or higher rate than their previous rate. They need to repeat the exercise until the mantra of breathing deeply and slowly works by reducing their pulse from the baselines of higher levels of anxiety (or distressed) states to a lower rate.”  

We also included discussion on gamification (third paragraph under section 10):

“In other studies by Li  J et al. (2014) and Fleming TM et al. (2017) , some participants suggested gamification strategies, which have shown promise in improving engagement with a digital platform interaction between users and providers. These strategies are associated with improvements in engagement and motivation to use digital health intervention.”

Figure 3: while it is stated later that there should be stakeholder engagement in software design and development, this is not brought out in figure 3.

We have included the stakeholders in Figure 3. We wrote: “This software development involves numerous stakeholders, namely: patients, therapists (psychiatrists and psychologists), and software developers.”

Section 7, line 232: the concept of affordance should be explained.

Thank you for the comment. We articulate the concept of affordance. We wrote:

“Affordance refers to the ability or role of an object to meet a certain action. It conceptualizes visual elements (or features) of the environment as information (new reference inserted: Gross, D., Stanney, K., & Cohn, J. (2005). Evoking affordances in virtual environments via sensory-stimuli substitution. Presence, 14(4), 482-491. doi>10.1162/105474605774785244). Affordance in VR is a pragmatic concept that allows the participants and the environment to interact to induce the flow state (i.e., subjectivity, perception, roles, context, and plays) in people. A passive user does not feel stimulated by the information but instead allows the person to take an active role in noticing and utilizing the information. Affordances are thus elements of the environment that enable action on the part of those who understand and recognize them.

Section 9: how will the respondents gauge their biological responses?

Thank you for the comment. We have included the articulation of the biological responses.

We wrote in the first paragraph of section 9:

"A distressed person typically experiences a higher level of pulse rate, respiration, and blood pressure. The blood pressure, pulse, and respiration will be monitored after each succession of psychological interventions. For example, in each session, a therapist will ask the respondents to perform a simple bio-feedback response. In this exercise, a directive in the form of a specific mantra," please breath slowly and deeply until you feel a sense of calmness," was immediately followed by the respondents’ re-checking their pulse after the relaxation technique in the immersive and interactive digital platform. Suppose the respondent has pulses at the same or higher rate than their previous rate. They need to repeat the exercise until the mantra of breathing deeply and slowly works by reducing their pulse from the baselines of higher levels of anxiety (or distressed) states to a lower rate."

Section 9, lines 282-285: why is it expensive? The involvement of the third party and the associated cost needs to be explained.

Thank you for your comment. We have added the discussion in the first paragraph of section 9:

"With third-party participation, i.e., with multiple avatars, we have to wait for other avatars to enter the session and engage within the online platform we are interacting with. Where time is a barrier to interacting with multiple avatars during immersion, this situation is regarded as expensive to run. Another example would be the involvement of numerous therapists (psychologists, psychiatrists, counselors, or trainers in psychological intervention) who need special training to monitor or conduct the session, which ultimately adds to the operational cost."

Figure 5: what was the nature of the implanted virtual reality tool? Was it mental health-specific?

Thank you for your comments. We try to search the term 'implanted,' but it is not mentioned anywhere in the text.

Section 10: there are no references to support any of the statements made.

 We have included some relevant references in Section 10 (Yang et al. 2021 and Kolbe et al. 2021).

Section 10: what is meant by "virtual reality arbitration". No reference is provided.

Thank you for your comments. We deleted the term and rephrase it in different sentences. In second last paragraph of section 10, we wrote:

“VR intervention is well-matched with the COVID-19 preventive plan, despite heightened procedures to avoid infection and maintain patient and provider safety.”

Reviewer 2 Report

Very interesting manuscript on use of VR as an intervention for psychosocial health during COVID. Although interesting, there are some small and large edits that need to be considered: 

Small edits:

Line 55 page 2: "Following this, the virtual reality (VR) experience in this kind of setting can be a stress reliever: - I would change this to "may" because using the word "can" sounds as though you have definitive proof via an experiment of best practice.

Remove the question - "what is stress?" The topic does not need to be initiated with a question and questions are not usually positied in scientific writing. Remove all of the questions in the headings and in the body of the paper. 

Rename the headings of the various sections so they are straightforward and clear. Currently, they are posed as questions, include semicolons as a 2-part title.... Remove all of that. For example - "Future Directions" is perfect for a heading title. "What is expected for the use of VR in specific populations with mental health problems" is not necessary or appropriate as a heading. 

Make sure the wording is not definitive in the absence of evidence with citations (will be vs. may be)

Decrease the amount of lay language and jargon throughout. 

Ensure the language throughout is non-stigmatizing - What is expected for the use of VR in specific populations with 315
mental health problems?      Instead consider: Implications for expanding the use of VR in communities experiencing COVID 19 mental health disparities. 

The example is Not for use as a heading since that is not an appropriate heading. The example is to show how to craft language in a non-stigmatizing way.

Large edits: 

The first paragraph needs to make the case for this narrative review. My first thought when I read narrative review was why is this not a systematic review.

Narrative reviews carry the lowest amount of rigor and are rarely cited because the rigor is low. Please substantiate why this is not a systematic review.

Also, was a scoping review considered since it has more rigor than a narrative review, but less than a systematic? Narrative reviews do not require any metrics to ensure a standardized approach - this must be addressed. 

It would be a good idea to provide stronger context to why individuals are experiencing mental health issues during COVID - provide global stats on the # infected and the # deceased. 

There is a lot of information in this manuscript and the flow is choppy. Early in the introduction section, there needs to be a description of what topics the reader will encounter. For example, "In this narrative review we will first introduce xyz. We will then describe xyz and lastly we will discuss....." 

Also, in the introduction section, in the last sentence the purpose of this manuscript needs to be stated - "The purpose of this article was to conduct a ________ review on the ......."  It should also be specified where the studies cited are based - are these UK studies? Canada studies? US studies? etc. Additionally, your sample needs to be specified - are these studies you are including going to focus on males/females? adults/adolescents/older adults? low income countries? etc. If this context is not provided, the read is confusing because the paragraphs describe young adults, healthcare workers, etc. This all needs to be structured so the read is not confusing.

In the next paragraph there needs to be justification for the reasons the type of review was used and how standardization and rigor was addressed. In narrative reviews its easy for selection bias of literature that will be discussed - so justification must be included. There needs to be more than 1 study cited per sentence to substantiate the claims in this manuscript. 

Throughout the rest of the paper definitions of uncommon words should be briefly defined. The studies should be synthesized with appropriate citations. 

There should be a limitations section that describes all of the limitation of the narrative review (if this remains a narrative review and doesn't upgrade to a scoping, integrative, or systematic) and also describe what measures were taken by the authors to mitigate some of the limitations. Additionally, there will always be a limitation to the barriers to VR technology. Although it is accessible - that does not mean that it is affordable and at reach for all persons. It would be remiss to not include this. VR is a luxury that is not available to all persons. 

In section 10 Future Direction, there is a study about hybrid rehab that is not cited at all. "VR acted as an escape for some...." There are no citations to this study in the entire paragraph. 

The conclusion section should be brief and summarize the purpose of the manuscript again and then summarize the findings and next steps concisely. 

Author Response

Reviewer.2

Very interesting manuscript on use of VR as an intervention for psychosocial health during COVID. Although interesting, there are some small and large edits that need to be considered: 

Small edits:

Line 55 page 2: "Following this, the virtual reality (VR) experience in this kind of setting can be a stress reliever: - I would change this to "may" because using the word "can" sounds as though you have definitive proof via an experiment of best practice.

We have changed the term from "can" to "may" in 2nd paragraph in p.2.

Remove the question - "what is stress?" The topic does not need to be initiated with a question and questions are not usually positied in scientific writing. Remove all of the questions in the headings and in the body of the paper. 

Thank you for your the recommendation. We have removed “What is stress?” in the section 2.

Rename the headings of the various sections so they are straightforward and clear. Currently, they are posed as questions, include semicolons as a 2-part title.... Remove all of that. For example - "Future Directions" is perfect for a heading title. "What is expected for the use of VR in specific populations with mental health problems" is not necessary or appropriate as a heading. 

Thank you for the suggestion.  We have renamed the title of section 10 as per suggestion by Reviewer 1.

Make sure the wording is not definitive in the absence of evidence with citations (will be vs. may be)

Thank you for your suggestions.

Decrease the amount of lay language and jargon throughout. 

Thank you for your comments. To our level best, we decreased the amount of lay language and jargon throughout. 

Ensure the language throughout is non-stigmatizing - What is expected for the use of VR in specific populations with 315
mental health problems?      Instead consider: Implications for expanding the use of VR in communities experiencing COVID 19 mental health disparities. 

Thank you for your suggestions. We have changed the title of section 10 as recommended.

The example is Not for use as a heading since that is not an appropriate heading. The example is to show how to craft language in a non-stigmatizing way.

Thank you and we took note of your suggestions.

Large edits: 

The first paragraph needs to make the case for this narrative review. My first thought when I read narrative review was why is this not a systematic review.

Thank you for your comments. Yes, indeed, we would like to search and study this pivotal area of research by exploring the topic of interest.

Narrative reviews carry the lowest amount of rigor and are rarely cited because the rigor is low. Please substantiate why this is not a systematic review.

We intend to embark on a systematic review when our intuitional board review allows us by giving their consent.

Also, was a scoping review considered since it has more rigor than a narrative review, but less than a systematic? Narrative reviews do not require any metrics to ensure a standardized approach - this must be addressed. 

Thank you for your comments. We have added few limitations to a narrative review. We included the discussion in a new subheading, just before the conclusion.

It would be a good idea to provide stronger context to why individuals are experiencing mental health issues during COVID - provide global stats on the # infected and the # deceased. 

There is a lot of information in this manuscript and the flow is choppy. Early in the introduction section, there needs to be a description of what topics the reader will encounter. For example, "In this narrative review we will first introduce xyz. We will then describe xyz and lastly we will discuss....." 

We have made the necessary changes and incorporated a study from a systematic review on the individuals who are experiencing mental health issues during COVID.

We wrote: “ In a systematic review published in 2020, a range of factors were associated with a higher risk of psychiatric sequelae and/or low mental well-being, including female gender, deteriorating self-related physical health, and families with COVID-19 (Reference: Vindegaard N, Benros ME. COVID-19 pandemic and mental health consequences: Systematic review of the current evidence. Brain Behav Immun. 2020;89:531-542. doi:10.1016/j.bbi.2020.05.048)”

Also, in the introduction section, in the last sentence the purpose of this manuscript needs to be stated - "The purpose of this article was to conduct a ________ review on the ......."  It should also be specified where the studies cited are based - are these UK studies? Canada studies? US studies? etc. Additionally, your sample needs to be specified - are these studies you are including going to focus on males/females? adults/adolescents/older adults? low income countries? etc. If this context is not provided, the read is confusing because the paragraphs describe young adults, healthcare workers, etc. This all needs to be structured so the read is not confusing.

Thank you for your comments. We have included the section by explaining the purpose of this review and structuring our review (last paragraph in section 2).

As this study is a narrative review, we are not able to state additional sample specification like gender preponderance, age group, and countries of studies, etc., as the study is unstructured and we have added the limitations of the review.

In the next paragraph there needs to be justification for the reasons the type of review was used and how standardization and rigor was addressed. In narrative reviews its easy for selection bias of literature that will be discussed - so justification must be included. There needs to be more than 1 study cited per sentence to substantiate the claims in this manuscript. 

We have added some limitations to a narrative review. We included the discussion in a new subheading, just prior to the conclusion.

Throughout the rest of the paper definitions of uncommon words should be briefly defined. The studies should be synthesized with appropriate citations. 

Thank you for your comments. We have defined some terms like “affordance” to make it clear to the readers. We also furnish relevant references to the newly added revisions (A total of 9 new references were added).

There should be a limitations section that describes all of the limitation of the narrative review (if this remains a narrative review and doesn't upgrade to a scoping, integrative, or systematic) and also describe what measures were taken by the authors to mitigate some of the limitations. Additionally, there will always be a limitation to the barriers to VR technology. Although it is accessible - that does not mean that it is affordable and at reach for all persons. It would be remiss to not include this. VR is a luxury that is not available to all people. 

Thank you for your comments. We have added a section on limitations to discuss this. In the second last paragraph of the writing, we wrote:

"11. Limitations

This review has limitations. A narrative approach often does not meet the essential criteria to mitigate bias. Not uncommonly, they lack unambiguous criteria for article selection. However, in our review, we included studies from a peer-reviewed journal, and thus, the credibility and validity of the studies were fair to good. We admitted a systematic review would fill the gap in the specialty of VR for psychological intervention among individuals with mental health problems.

On another point, there will always be a limit to the barriers to VR technology. Although it is accessible, that does not mean that it is affordable and reachable for all users. In other words, VR is a luxury that is not available to all people."

In section 10 Future Direction, there is a study about hybrid rehab that is not cited at all. "VR acted as an escape for some...." There are no citations to this study in the entire paragraph. 

Thank you for your comments. We have included the reference for Kolbe et al. (2021).

Kolbe L, Jaywant A, Gupta A, Vanderlind WM, Jabbour G. Use of virtual reality in the inpatient rehabilitation of COVID-19 patients. Gen Hosp Psychiatry. 2021;71:76-81. doi:10.1016/j.genhosppsych.2021.04.008

The conclusion section should be brief and summarize the purpose of the manuscript again and then summarize the findings and next steps concisely. 

Thank you for your comments and we are doing our level best for the conclusion. In conclusion, we wrote:

“…. , where patients experienced positive psychological effects, especially in a naturalistic setting and in control of their bodily symptoms”.

Reviewer 3 Report

Dear Authors, 

Thank you for the opportunity to review the paper entitled “The Role of Virtual Reality and Software Development as a Psychological Intervention for Mental Health Disturbances during COVID-19 Pandemic: A Narrative Review”. The paper presents itself as coherent, identifying particular areas of VR use with psychological issues. I have no major concerns about the paper. It just seems to me that Chapter 9 should also include information about garden-based interventions - there is a documented position in many papers that this very setting is beneficial for most patients. Please see:

  • Riva et al. COVID Feel Good-An Easy Self-Help Virtual Reality Protocol to Overcome the Psychological Burden of Coronavirus. Front Psychiatry. 2020 Sep 23;11:563319
  • Rutkowski et al. Evaluation of the Efficacy of Immersive Virtual Reality Therapy as a Method Supporting Pulmonary Rehabilitation: A Randomized Controlled Trial. J. Clin. Med. 2021, 10(2), 352
  • Alyan et al. The Influence of Virtual Forest Walk on Physiological and Psychological Responses. Int. J. Environ. Res. Public Health 2021, 18(21), 11420;

I think it's worth integrating this information. Congratulations, on your interesting study.

Author Response

Reviewer.3

Dear Authors, 

Thank you for the opportunity to review the paper entitled "The Role of Virtual Reality and Software Development as a Psychological Intervention for Mental Health Disturbances during COVID-19 Pandemic: A Narrative Review". The paper presents itself as coherent, identifying particular areas of VR use with psychological issues. I have no major concerns about the paper. It just seems to me that Chapter 9 should also include information about garden-based interventions - there is a documented position in many papers that this very setting is beneficial for most patients. Please see:

  • Riva et al. COVID Feel Good-An Easy Self-Help Virtual Reality Protocol to Overcome the Psychological Burden of Coronavirus. Front Psychiatry. 2020 Sep 23;11:563319
  • Rutkowski et al. Evaluation of the Efficacy of Immersive Virtual Reality Therapy as a Method Supporting Pulmonary Rehabilitation: A Randomized Controlled Trial. J. Clin. Med. 2021, 10(2), 352
  • Alyan et al. The Influence of Virtual Forest Walk on Physiological and Psychological Responses. Int. J. Environ. Res. Public Health 2021, 18(21), 11420;

Thank you for your comments. We have included al THREE references and also discussed these suggestions in section 9. We wrote:

“An anxious or distressed person typically experiences a higher level of pulse rate, respiration, and blood pressure. The blood pressure, pulse, and respiration will be monitored after each succession of psychological interventions. For example, in each session, a therapist will ask the respondents to perform a simple bio-feedback response. In this exercise, a directive in the form of a specific mantra," please breath slowly and deeply until you feel a sense of calmness," was immediately followed by the respondents’ rechecking of their pulse after the relaxation technique in the immersive and inter-active digital platform. Suppose the respondent has the pulses at the same or higher rate than their previous rate. They need to repeat the exercise until the mantra of breathing deeply and slowly works by reducing their pulse from the baseline of higher levels of anxiety (or distressed) states to a lower rate.  

Based on a protocol developed by Riva et al (2020) [57], instituting a natural setting environment like a secret garden would be another emotionally-inspiring strategy to help psychologically distressed people. The Secret Garden is a 360-degree VR scenario where participants are engaged in a naturalistic setting and be in a “safe haven” in a digital platform. They felt calm and relaxed, which is far from the stressful circumstances they experienced in their routine everyday contexts. Gradually, they may learn how to control their distressing feelings and reflect on their experiences in a guided protocol. In another study by Alyan et al (2021), a participant who was exposed to stimuli derived from greenery scenarios in virtual forest environments showed beneficial stress-relieving effects [58]. This is based on the outcomes of two physiological techniques that determine their heart rate and skin conductance levels. The beneficial effects would be profound if realistic graphics were to be used. This exposure contributes to the concept of forest therapy and provides new directions for future forest therapy research. For the role of immersive VR therapy in relieving stress symptoms among people with a physical disorder like chronic obstructive airway disease, Rutkowski et al. (2021) reported benefits in terms of the mood improvement role among patients with lung disease” [59].

References added in the references section:

Riva, Giuseppe & Bernardelli, Luca & Browning, Matthew & Castelnuovo, Gianluca & Cavedoni, Silvia & Chirico, Alice & Cipresso, Pietro & Bengel de Paula, Dirce Maria & Di Lernia, Daniele & Figueras-Puigderrajols, Natàlia & Fernán-dez-Álvarez, Javier & Fuji, Kei & Gaggioli, Andrea & Gutiérrez-Maldonado, José & Hong, Upyong & Mancuso, Valentina & Mazzeo, Milena & Molinari, Enrico & Moretti, Luciana & Wiederhold, Brenda. (2020). COVID Feel Good – An Easy Self-Help Virtual Reality Protocol to Overcome the Psychological Burden of Coronavirus. 10.31234/osf.io/6umvn.

Rutkowski S, Szczegielniak J, Szczepańska-Gieracha J. Evaluation of the Efficacy of Immersive Virtual Reality Therapy as a Method Supporting Pulmonary Rehabilita-tion: A Randomized Controlled Trial. J Clin Med. 2021 Jan 18;10(2):352. doi: 10.3390/jcm10020352. PMID: 33477733; PMCID: PMC7832322.

Alyan E, Combe T, Awang Rambli DR, Sulaiman S, Merienne F, Diyana N. The In-fluence of Virtual Forest Walk on Physiological and Psychological Responses. Int J En-viron Res Public Health. 2021;18(21):11420. Published 2021 Oct 29. doi:10.3390/ijerph182111420

I think it's worth integrating this information. Congratulations, on your interesting study.

Thank you for your generous comments.

Round 2

Reviewer 1 Report

Thank you for the opportunity to review the revised paper. Most of the queries have been resolved, but some of the responses raise further queries. 

Previous query:

A problem with narrative reviews is that the authors are not required to report their search strategy, inclusion and exclusion criteria, or the databases searched. This makes the process unreproducible and introduces the risk of author bias.

Response

We thank the reviewer for the comment. We are aware that narrative reviews are unreproducible and risk author bias, but we have included our citations. We also addressed the limitation in a new subheading just prior to the conclusion. We wrote:

"11. Limitations

This review has a few limitations. A narrative approach often does not meet the essential criteria to mitigate bias. Not uncommonly, they lack unambiguous criteria for article selection. However, in our review, we included studies from a peer-reviewed journal, and thus, the credibility and validity of the studies were fair to good. We admitted a systematic review would fill the gap in the specialty of VR for psychological intervention among individuals with mental health problems. On another point, there will always be a limit to the barriers to VR technology. Although it is accessible, that does not mean that it is affordable and reachable for all users. In other words, VR is a luxury that is not available to all people."

New query

Line 433: “we included studies from a peer-reviewed journal” You have included studies from a number of journals.

The fact that you used peer-reviewed literature does not mitigate bias. There is selection and author bias, which is why the databases and search criteria should be stated.

Previous query

It is not clear if the authors have developed software, line 251, or if they have developed and used software 298-304 as there is no reference cited for the "current study".

Response

We refer to a "current study" as "a near-future study." We intend to develop the software as part of the team's Ph.D. program. In the next two years, we want to build a software program for psychological intervention among patients with mental health problems, and the first author (M.H.H.) is currently doing his Master's in Medical Science, which later will be converted to a Ph.D.

We re-structure the sentences in the same paragraph as: "The introduction of this innovation into a hybrid medicine-rehabilitation inpatient unit resulted in a highly positive reaction in a near-future study."

New query

The new sentence is poorly constructed. If the software has not yet been developed and e study has not been undertaken, how can there have been “a highly positive reaction in a near-future study”?

Previous query 

Section 3, lines 93-97: avoiding face-to-face intervention may be best during the Covid-19 pandemic, but the implication is that virtual reality is the best intervention. No data are presented to suggest that it is better than a videoconference intervention with a psychiatrist or psychologist.

Response

Thank you for bringing up this issue. As stated in reference no. 46, and in the text (second last paragraph in section 6), we wrote: " Recorded 360° video education in meditation viewed through an HMD (VR/3D format) appears to provide a considerable experiential benefit over teachings presented in 2D design and may provide a safe—and for some, even preferred—alternative to teaching meditation face-to-face." This study by Waller et al. (2021) clearly states that the 3-D format communication, which is the fundamental concept of VR (i.e., being immersive and interactive), is more favourable in meditation teaching than the 2-D format. Videoconference intervention with the psychiatrist or psychologist lacks immersion and self-empowerment to conduct the desirable therapeutical sessions at their convenient time. Also, in a videoconference, the interaction time is minimal due to constraints like limited usability (as a digital platform like Zoom or GoogleMeet has its own terms and conditions).

New query

The response is only relevant for meditation. There is more to psychological interventions for mental health disturbances than meditation? Is VR relevant for other methods?

Previous query

Section 9: how will the respondents gauge their biological responses?

Response

Thank you for the comment. We have included the articulation of the biological responses.

We wrote in the first paragraph of section 9:

"A distressed person typically experiences a higher level of pulse rate, respiration, and blood pressure. The blood pressure, pulse, and respiration will be monitored after each succession of psychological interventions. For example, in each session, a therapist will ask the respondents to perform a simple bio-feedback response. In this exercise, a directive in the form of a specific mantra," please breath slowly and deeply until you feel a sense of calmness," was immediately followed by the respondents’ re-checking their pulse after the relaxation technique in the immersive and interactive digital platform. Suppose the respondent has pulses at the same or higher rate than their previous rate. They need to repeat the exercise until the mantra of breathing deeply and slowly works by reducing their pulse from the baselines of higher levels of anxiety (or distressed) states to a lower rate."

New query 

The response only addresses pulse rate. How will the blood pressure be monitored? 

Previous query

Figure 5: what was the nature of the implanted virtual reality tool? Was it mental health-specific?

I apologise, the word “implanted” should have been “implemented”

New comments

The response have some grammatical and layout errors. For example, lines 102 , 314, 394.

Reviewer 2 Report

Interesting manuscript on use of VR as an intervention for psychosocial health during COVID. Revisions suggested for both large and small edits were not fully considered based on the revised manuscript.

The biggest concern from the fist review and now is the lack of description and/or rationale for not conducting a systematic review. It is not common for a systematic review to require IRB approval since human subjects are not involved. A systematic review is an abstraction of indexed data – this does not require approvals. This paper is a narrative review – this is the lowest form of review possible and subject to extensive bias in the selection of articles and also in the quality of those articles that are selected. The authors stated in the paper that the although a narrative review is not the gold standard, the articles selected in this manuscript were peer reviewed – as to convey ALL peer reviewed work is of good quality. This is unacceptable.

The expectation was the authors would simply state something like: It was not possible to conduct a systematic review in this area bc…..(theres not extensive evidence available….xyz…). They could then in the limitations state that the narrative is the weakest form of review AND to take the findings with caution withstanding that the data is still an incremental step in the field for this topic.

The authors should have really conducted a scoping review, which would have been most appropriate for manuscripts on topics with little or not enough evidence for a systematic review.

Additionally, in ANY review manuscript there is always a research question that guides the search. So the omission of any characteristics of studies included does not make any sense. The papers that were selected for the narrative review should have included a table with aggregated information on descriptive characteristics at minimum.

Because of this, I am unable to state that the manuscript has been critically revised to a publishable standard.
